# Recruitment and positioning determine the specific role of the XPF-ERCC1 endonuclease in interstrand crosslink repair

Daisy Klein Douwel, Wouter S Hoogenboom, Rick ACM Boonen[†] & Puck Knipscheer[*] 🆔

## Abstract

XPF-ERCC1 is a structure-specific endonuclease pivotal for several DNA repair pathways and, when mutated, can cause multiple diseases. Although the disease-specific mutations are thought to affect different DNA repair pathways, the molecular basis for this is unknown. Here we examine the function of XPF-ERCC1 in DNA interstrand crosslink (ICL) repair. We used *Xenopus* egg extracts to measure both ICL and nucleotide excision repair, and we identified mutations that are specifically defective in ICL repair. One of these separation-of-function mutations resides in the helicase-like domain of XPF and disrupts binding to SLX4 and recruitment to the ICL. A small deletion in the same domain supports recruitment of XPF to the ICL, but inhibited the unhooking incisions most likely by disrupting a second, transient interaction with SLX4. Finally, mutation of residues in the nuclease domain did not affect localization of XPF-ERCC1 to the ICL but did prevent incisions on the ICL substrate. Our data support a model in which the ICL repair-specific function of XPF-ERCC1 is dependent on recruitment, positioning and substrate recognition.

**Keywords** Fanconi anemia; interstrand crosslink repair; nucleotide excision repair; *Xenopus* egg extract; XPF-ERCC1

**Subject Categories** DNA Replication, Repair & Recombination; Molecular Biology of Disease

The EMBO Journal (2017) 36: 2034–2046

See also: **UB Abdullah *et al*** (July 2017) and **OD Schärer** (July 2017)

## Introduction

The structure-specific endonuclease XPF-ERCC1 participates in multiple genome maintenance pathways, including nucleotide excision repair (NER), DNA interstrand crosslink (ICL) repair, certain branches of double-stranded break (DSB) repair, and telomere maintenance. Mutations in XPF-ERCC1 have been associated with the

genetic disorders Xeroderma pigmentosum (XP), Cockayne syndrome (CS), cerebro-oculo-facio-skeletal syndrome (COFS), Fanconi anemia (FA), and premature aging. These phenotypes are believed to be caused by a defect in one, or several, of the genome maintenance pathways XPF-ERCC1 is involved in, but the molecular basis for this is unknown.

XPF is a structure-specific endonuclease that contains an N-terminal helicase-like domain, a central ERCC4-type nuclease domain, and a C-terminal helix-hairpin-helix (HhH) domain with which it interacts with its cofactor ERCC1 (Fig 1A). Very little is known about the role of the helicase-like domain, but it is important for nuclease activity (Bowles *et al*, 2012). The function of XPF-ERCC1 in NER, a pathway that removes helix distorting lesions, has been extensively studied (Gillet & Schärer, 2006; Friedberg, 2011). After damage recognition by upstream NER factors, XPF-ERCC1 is recruited to the lesion by XPA and excises a short oligo containing the damage in collaboration with another nuclease XPG (Huang *et al*, 1992; Li *et al*, 1995; Spivak, 2015). Defects in NER factors are associated with the genetic disease Xeroderma pigmentosum (XP), which is characterized by sunlight sensitivity and skin cancer predisposition. Uniquely among NER factors, deficiency in XPF-ERCC1 not only causes UV sensitivity, but also results in hypersensitivity to ICL-inducing agents, indicating an additional role for this protein in the repair of interstrand crosslinks (De Silva *et al*, 2000; Kuraoka *et al*, 2000; Niedernhofer *et al*, 2004).

ICLs are toxic DNA lesions that covalently link both strands of the DNA together, thereby blocking DNA replication and transcription. ICLs are formed endogenously by products of cellular metabolism, but are also induced at high doses by certain chemotherapeutic drugs. The major pathway of ICL repair is coupled to DNA replication and involves the coordinated action of many DNA repair proteins including the Fanconi anemia (FA) pathway proteins. Mutations in any of the 21 currently known FA genes give rise to Fanconi anemia (FA), a cancer susceptibility disorder characterized by cellular sensitivity to ICL-inducing agents (Kottemann & Smogorzewska, 2013; Dong *et al*, 2015). Using a *Xenopus* egg extract-based assay, we and others have recently elucidated a molecular mechanism of replication-coupled ICL repair (Fig EV1; Räschle *et al*, 2008). This mechanism requires the convergence of

Hubrecht Institute - KNAW, University Medical Center Utrecht & Cancer GenomiCs Netherlands, Utrecht, The Netherlands
*Corresponding author. Tel: +31 302121800; E-mail: p.knipscheer@hubrecht.eu
†Present address: Department of Human Genetics, Leiden University Medical Center, Leiden, The Netherlands

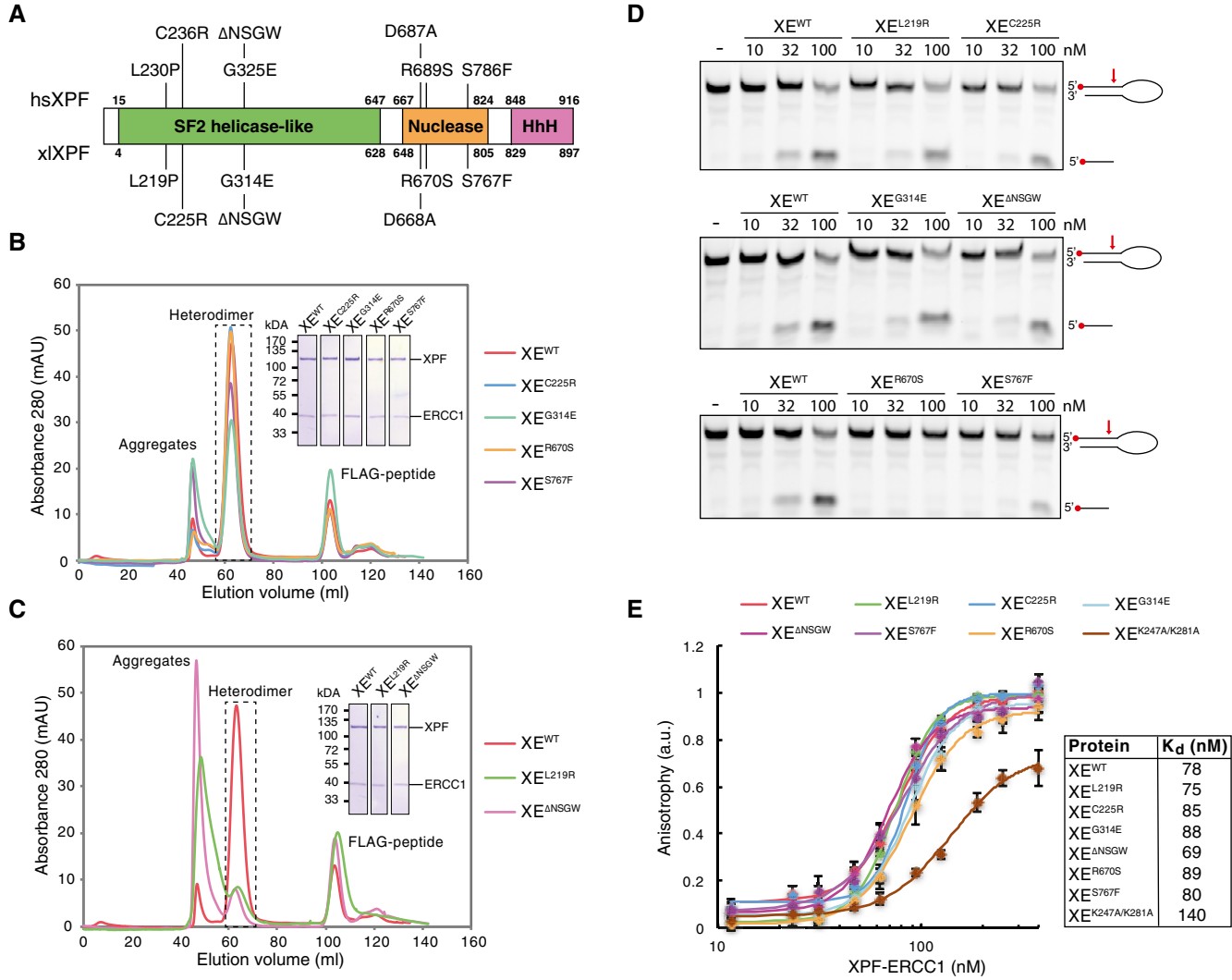

**Figure 1. Characterization of mutant XPF-ERCC1 complexes.**

A  Schematic representation of the domain organization of the XPF protein. Domain boundaries of human and *Xenopus laevis* XPF are indicated. Relevant mutations of the human protein, and the *Xenopus laevis* equivalents, are indicated on top and bottom, respectively.

B  Superdex 200 gel filtration column elution profile of wild-type XPF-ERCC1 and indicated mutant complexes. Aggregates eluted in the void volume of the column (~45 ml) while the active XPF-ERCC1 heterodimer eluted at ~65 ml. The peak eluting at ~105 ml contains the FLAG peptide used to elute the protein from the FLAG affinity resin. The heterodimer peak was isolated, and proteins were separated on SDS–PAGE and stained with Coomassie blue (inset).

C  As in (B) but for different mutant complexes that showed more aggregation.

D  Wild-type and indicated mutant XPF-ERCC1 complexes were incubated with a 5′-FAM-labeled stem-loop DNA substrate (10 nM) at room temperature for 30 min. Reaction products were separated on a 12% urea–PAGE gel and visualized using a fluorescence imaging system. Red arrow indicates position of incision by XPF-ERCC1.

E  Wild-type and mutant XPF-ERCC1 complexes at various concentrations were incubated with a 5′-FAM-labeled 3′ flap DNA substrate (10 nM) and fluorescent anisotropy was measured. Graphs were fitted to calculate dissociation constants ($K_d$s) as described in the Materials and Methods section. The error bars represent s.d. from three measurements. Experimental replicates are shown in Fig EV2.

Source data are available online for this figure.

two replication forks at an ICL during DNA replication (Zhang *et al*, 2015). Both forks initially stall 20–40 nucleotides from the crosslink followed by CMG helicase unloading allowing one fork to approach to within 1 nucleotide of the crosslink (Räschle *et al*, 2008; Fu *et al*, 2011; Long *et al*, 2011). Dual incisions on either side of the ICL then unhook the lesion from one of the strands. This critical repair step requires the endonuclease XPF (FANCQ)-ERCC1, which is recruited to the ICL by the large scaffold protein SLX4 (FANCP), and depends

on the activation of the Fanconi anemia pathway by ubiquitylation of the FANCI-FANCD2 complex (Knipscheer *et al*, 2009; Klein Douwel *et al*, 2014). After unhooking, a nucleotide is inserted across from the adducted base, followed by strand extension by REV1 and polymerase ζ, consisting of REV7 (FANCV) and REV3 (Räschle *et al*, 2008; Budzowska *et al*, 2015; Mamrak *et al*, 2016). This strand now acts as a template for repair of the opposite strand by HR (Long *et al*, 2011), leading to fully repaired products.

Patient phenotypes linked to specific XPF mutations can be extremely valuable in determining pathway-specific functions. Most patients with a mutation in XPF suffer from a mild form of XP and are deficient in NER. These patients express residual protein and are likely proficient in ICL repair, because they do not show features of FA (Ahmad *et al*, 2010). In some cases, XPF mutations can lead to much more severe phenotypes. An extreme progeroid syndrome was caused by a mutation in the helicase-like domain of XPF (R153P). This patient suffered from neurological and hematological defects and a cellular sensitivity to UV and ICLs indicating both NER and ICL repair were defective (Niedernhofer *et al*, 2006). Another patient, with a mutation in the same XPF domain (C236R), presented with phenotypes of XP, but also of CS, such as developmental and neurological abnormalities (Kashiyama *et al*, 2013). This patient also showed FA-like features and ICL sensitivity suggestive of a defect in ICL repair. In addition, some patients with specific mutations in XPF were diagnosed with Fanconi anemia and showed no signs of XP (Bogliolo *et al*, 2013). These mutations were mapped to the helicase-like domain (L230P), and the nuclease domain (R689S) of XPF.

To examine what features of XPF-ERCC1 determine its specificity in ICL repair, we employed the *Xenopus* egg extract system. We monitored both replication-coupled ICL repair and nucleotide excision repair and identified five XPF mutants that are deficient in ICL repair and proficient in NER. Although all of these mutants showed a defect in ICL unhooking, the majority was still efficiently recruited to the ICL. In contrast, mutation of *xl*XPF leucine 219, equivalent to the human leucine 230 mutated in Fanconi anemia, abrogated this ICL localization. This was caused by a defect in interaction with SLX4. We propose there are two interaction sites between XPF and SLX4, one ensuring recruitment of XPF to the ICL, and another to promote its nuclease activity. This dual interaction site, in combination with residues in the nuclease domain ensuring substrate specificity, dictates the ICL repair-specific function of XPF-ERCC1.

# Results

## XPF mutants form functional complexes with ERCC1

To study the role of XPF-ERCC1 in ICL repair, we selected a set of XPF mutations that we predicted to specifically affect this process. In the helicase-like domain, two point mutations were found in patients with FA and FA-like symptoms, L230P and C236R (Bogliolo *et al*, 2013; Kashiyama *et al*, 2013). These mutated residues correspond to residues L219 and C225 in *Xenopus laevis* XPF that is 75% identical to human XPF (Fig 1A). Another mutation in XPF's helicase-like domain, *hs*G325E (*xl*G314E), was reported to disrupt the interaction of XPF with the BTB domain of SLX4 (Andersen *et al*, 2009). This interaction is likely specific to ICL repair and not NER because SLX4-deficient cells are not UV sensitive (Crossan *et al*, 2011). With the aim to further disrupt this interaction, we generated a deletion mutant lacking G314 and three surrounding residues that were predicted to form a loop (XPF$^{\Delta NSGW}$). Finally, in the nuclease domain, we analyzed two mutations, R689S and S786F (*xl*R670S and *xl*S676F). The R689S mutation was associated with Fanconi anemia (Bogliolo *et al*, 2013) and the S786F mutation sensitizes cells to MMC, but not UV radiation (Osorio *et al*, 2013).

We co-expressed FLAG-tagged *xl*XPF wild type and mutants with His-tagged *hs*ERCC1 in Sf9 insect cells and purified the complex by affinity purification. We previously showed that the *xl*XPF-*hs*ERCC1 complex (referred to as XPF-ERCC1 from here on) supports ICL repair in *Xenopus* egg extracts (Klein Douwel *et al*, 2014). Expression levels of all mutant complexes were similar to wild-type levels except for the XPF$^{L219P}$-ERCC1 (XE$^{L219P}$) complex that was previously reported to be unstable (Bogliolo *et al*, 2013; Hashimoto *et al*, 2015). To examine the function of leucine 219, we instead mutated it to an arginine and found that XE$^{L219R}$ was expressed at normal levels. When wild-type XPF-ERCC1 was subjected to gel filtration chromatography, the majority of the protein eluted at the expected range for a heterodimer while a small fraction of inactive aggregates eluted in the void volume of the column, similar to what was previously described (Fig 1B; Enzlin & Schärer, 2002). The XE$^{C225R}$, XE$^{G314E}$, XE$^{R670S}$, and XE$^{S767F}$ mutant complexes behaved similarly to the wild type on gel filtration (Fig 1B) and the peak containing the heterodimer was isolated and used for further experiments. However, the XE$^{L219R}$ and XE$^{\Delta NSGW}$ mutant complexes showed an increased aggregate peak and lower heterodimer peak (Fig 1C). Nevertheless, when this heterodimer peak was isolated and rerun on a gel filtration column, it did not aggregate (Fig EV2A).

We next examined the endonuclease activity of the mutant XPF-ERCC1 complexes. To this end, a fluorescently labeled stem-loop substrate was incubated with increasing concentrations of protein, and reaction products were separated by denaturing urea–PAGE. All XPF-ERCC1 complexes with mutations in the helicase-like domain showed nuclease activity similar to wild-type protein, as seen by the appearance of the incision product (Fig 1D, top two panels, and Fig EV2B). Importantly, this demonstrates that the XE$^{L219R}$ and XE$^{\Delta NSGW}$ complexes, which showed increased aggregation upon expression, are fully active after isolation of the heterodimer peak. The XE$^{S767F}$ complex was slightly reduced in nuclease activity while the XE$^{R670S}$ complex showed a more dramatic reduction and was only capable of cutting the substrate at high concentrations (Figs 1D and EV2D). This was not surprising as both mutations are located in the nuclease domain of XPF, and the human equivalent of the R670S mutant has decreased nuclease activity (Enzlin & Schärer, 2002; Su *et al*, 2012). We also analyzed the endonuclease activity on a 3′ flap substrate and obtained similar results (Fig EV2C). In conclusion, all mutants except R670S have nuclease activity similar to wild-type protein.

Finally, we analyzed the DNA binding of the mutant XPF-ERCC1 complexes. For this, a fluorescently labeled 3′ flap substrate was incubated with increasing concentrations of XPF-ERCC1 and fluorescence anisotropy was measured to assess binding. All our mutants showed very similar binding curves and the $K_d$ values derived from these curves were comparable to each other and to wild-type XPF-ERCC1 (Figs 1E and EV2E). This indicates that the mutations do not affect DNA binding affinity. To validate our results, we measured fluorescence anisotropy of a mutant XPF-ERCC1 carrying two point mutations in ERCC1 (K247A and K281A) that were previously shown to affect DNA binding (Su *et al*, 2012). Consistent with this, we found that this XE$^{KAKA}$ mutant had reduced affinity for DNA (Figs 1E and EV2E).

In summary, we purified six mutant XPF-ERCC1 complexes that are predicted to affect ICL repair. All mutant complexes form stable heterodimers and interact normally with DNA. The nuclease activity

of the mutants is similar to wild type, with the exception of one mutation in the nuclease domain.

## Mutations in helicase-like and nuclease domains abrogate ICL repair

To investigate the effect of the XPF mutations on ICL repair, we used *Xenopus* egg extracts. This system recapitulates DNA replication-coupled repair of a sequence-specific cisplatin ICL situated on a plasmid template (pICL; Räschle *et al*, 2008). Moreover, it enables the quantification of repair by the regeneration of a SapI restriction site that is blocked by the crosslink (Fig 2A). We immunodepleted ERCC1 from egg extract and complemented the repair reaction with wild-type or mutant XPF-ERCC1. Since depletion of ERCC1 leads to equal depletion of XPF (Klein Douwel *et al*, 2014), we refer to this depletion as an XPF-ERCC1 depletion. Because depletion of

XPF-ERCC1 leads to co-depletion of SLX4, we also complemented all the depleted reactions with purified *xl*SLX4 protein unless stated otherwise (Fig EV3A; Klein Douwel *et al*, 2014). Reactions were stopped at various time points, and DNA repair intermediates were isolated and digested with SapI to quantify ICL repair. A small fraction of non-crosslinked plasmids is present in pICL preparations leading to a constant background of SapI digestible replication products. XPF-ERCC1-depleted extracts did not support ICL repair above this background while addition of recombinant wild-type XPF-ERCC1 (XE$^{wt}$) restored ICL repair (Fig 2B–D; Klein Douwel *et al*, 2014). XE$^{L219R}$ and XE$^{C225R}$, which carry mutations in the helicase-like domain of XPF, did not efficiently rescue ICL repair (Figs 2B and EV3B). We then tested the other two helicase-like domain mutants that were expected to affect the interaction with the BTB domain of SLX4. While addition of XE$^{G314E}$ to XPF-ERCC1-depleted extract supported ICL repair, the deletion mutant XE$^{\Delta NSGW}$ was

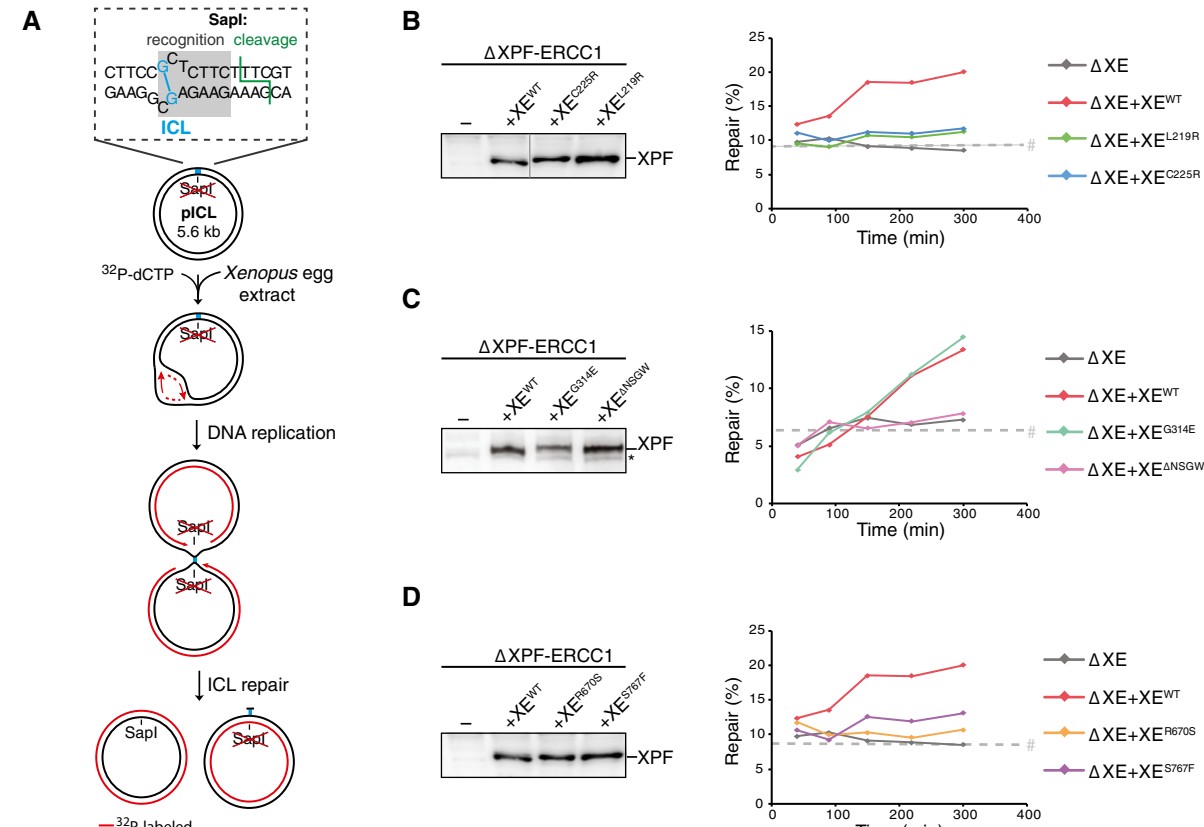

**Figure 2.  Effect of mutations in XPF-ERCC1 on ICL repair in *Xenopus* egg extract.**

A    Schematic representation of repair of a plasmid containing a site-specific cisplatin ICL (pICL) in *Xenopus* egg extract. The SapI site that is blocked by the ICL becomes available on one of the replicated molecules after full repair via HR using the sister molecule (Fig EV1). The sister molecule is repaired by lesion bypass, but retains the unhooked ICL that is not removed efficiently in *Xenopus* egg extract (Räschle *et al*, 2008).

B    XPF-ERCC1-depleted (ΔXE) and XPF-ERCC1-depleted extracts complemented with wild-type (XE$^{WT}$) or indicated mutant XPF-ERCC1 (XE$^{MUT}$) were analyzed by Western blot using α-XPF antibodies (left panel). Line within blot indicates position where irrelevant lanes were removed. These extracts were used to replicate pICL. Replication intermediates were isolated and digested with HincII, or HincII and SapI, and separated on agarose gel. Repair efficiency, represented by SapI regeneration, was calculated as described (Räschle *et al*, 2008) and plotted (right panel).

C, D  As in (B) but analyzing different XPF-ERCC1 mutant complexes. Experimental replicates are shown in Fig EV3. Note: repair levels can differ per batch of individually prepared extract or per depletion experiment and can only be compared within an experiment. *, background band.

Data information: (B–D) #, SapI fragments from contaminating uncrosslinked plasmid present in varying degrees in pICL preparations.
Source data are available online for this figure.

defective in ICL repair (Figs 2C and EV3C). This suggests that this region is important for ICL repair, likely through mediating interaction with SLX4. Finally, XPF-ERCC1-depleted extracts were supplemented with the nuclease domain mutants XE$^{R670S}$ and XE$^{S767F}$. Both mutants were unable to restore ICL repair (Figs 2D and EV3B). These results show that specific residues in the helicase-like domain and the nuclease domain of XPF-ERCC1 are required for ICL repair.

### ICL repair-deficient XPF mutants are proficient in NER

To determine whether the XPF mutations specifically affect ICL repair, we investigated their activity in nucleotide excision repair (NER). During NER, the endonucleases XPF-ERCC1 and XPG make incisions on either side of a lesion, creating a gap that is subsequently filled in. This gap-filling DNA synthesis is called unscheduled DNA synthesis (UDS) to differentiate it from the semi-conservative DNA synthesis that takes place during replication. UDS can be measured on UV-damaged plasmids incubated in a high-speed supernatant *Xenopus* egg extract and used as a readout for NER activity (Fig 3A; Shivji *et al*, 1994; Gaillard *et al*, 1996). To this end, we incubated non-damaged or UV-damaged plasmid in a non-replicating *Xenopus* egg extract in the presence of $^{32}$P-dCTP. The DNA was subsequently isolated and linearized, and the products were separated on an agarose gel (Fig 3B). While a UV-damaged plasmid showed clear incorporation of $^{32}$P-dCTP indicative of UDS, a non-damaged plasmid only showed some background incorporation, probably due to nicks created during plasmid preparation (Fig 3B, lanes 1 and 2). To confirm that the UDS on the UV-damaged template

is a result of the repair, we directly monitored cyclobutane pyrimidine dimers and showed that *Xenopus* egg extract is capable of removing these lesions (Fig EV4C). To further validate that the unscheduled DNA synthesis is caused by NER, we showed that UDS is strongly reduced after depletion of NER factors PCNA as well as XPA (Fig EV4D and E). In addition, we found that immunodepletion of XPF (Fig EV4A) strongly reduced UV-dependent UDS (Fig 3B, compare lanes 2 and 4, and Fig 3C). The slight increase in UDS compared to the non-damaged plasmid is likely caused by an incomplete depletion of XPF-ERCC1 or other repair mechanisms present in the extract. Addition of wild-type XPF-ERCC1 (XE$^{WT}$) to XPF-depleted extracts fully rescued UDS, while addition of a catalytically inactive XE$^{D668A}$ mutant did not support UDS (Fig 3B, compare lanes 5 and 10, Figs 3C and EV4B). This shows that XPF-ERCC1 is required for UDS in *Xenopus* egg extract. We then complemented an XPF-ERCC1-depleted extract with the XPF-ERCC1 mutants and found that all mutants were able to rescue the NER defect (Fig 3B and C). This observation was especially striking for the XE$^{R670S}$ mutant whose nuclease activity on model DNA templates was strongly reduced (Fig 1D). This finding is consistent with a previous report in which the human equivalent of this mutant was able to make NER incisions, although there was a difference in the position of the incisions compared to the wild-type protein (Su *et al*, 2012).

In summary, we identified three mutations in the helicase-like domain (XE$^{L219R}$, XE$^{C225R}$ and XE$^{\Delta NSGW}$) and two in the nuclease domain (XE$^{R670S}$ and XE$^{S767F}$) that are defective in ICL repair, while supporting functional NER. Strikingly, this separation of function is achieved by mutations in two different domains of XPF-ERCC1.

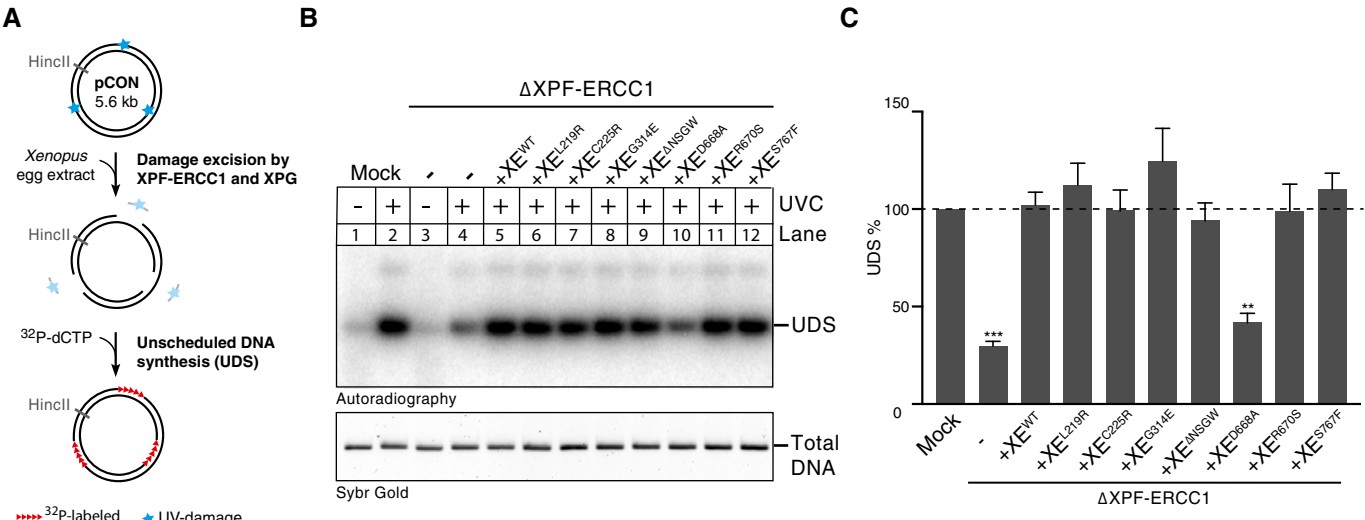

**Figure 3. XPF-ERCC1 mutant complexes are active in nucleotide excision repair (NER).**

A  Schematic representation of unscheduled DNA synthesis (UDS) during NER on a UV-treated template in high-speed supernatant (HSS) egg extract.

B  Mock-depleted, XPF-ERCC1-depleted (ΔXE), and XPF-ERCC1-depleted extracts complemented with wild-type (XE$^{WT}$) or mutant XPF-ERCC1 (XE$^{MUT}$) were incubated with untreated or UV-treated plasmids for 2 h at room temperature in the presence of $^{32}$P-α-dCTP. Reaction products were isolated, linearized with HincII, and separated on a 0.8% agarose gel. The DNA was visualized by autoradiography to show incorporation of $^{32}$P-α-dCTP during UDS (upper panel) and stained with SYBR gold for total DNA (lower panel).

C  The incorporation of $^{32}$P-α-dCTP was quantified, the background signal from non-damaged plasmid was subtracted, and the signal for the mock depletion condition was set to 100% to normalize the data. Error bars represent s.e.m. of three independent experiments. **$P = 0.003$, ***$P = 0.0004$, paired *t*-test comparing all conditions to the mock. All non-marked conditions did not show a statistical difference from the mock condition.

Source data are available online for this figure.

## Separation-of-function mutations in XPF specifically affect ICL incisions

To determine the mechanism underlying the specific inhibition of ICL repair in these mutants, we examined which step in ICL repair was affected. We previously showed that XPF is required for ICL unhooking (Klein Douwel *et al*, 2014). However, XPF could also have additional roles downstream, for example, in HR, that might be affected by our mutations (Bergstralh & Sekelsky, 2008). To directly monitor unhooking incisions that take place on the parental strand, we pre-labeled pICL with $^{32}$P-dCTP using nick translation and replicated it in *Xenopus* egg extract. Replication intermediates were linearized and separated on a denaturing agarose gel. At early times, the parental strand migrates as a large X-structure, while after crosslink unhooking during repair, it is converted to a linear molecule and arms (Fig 4A; Knipscheer *et al*, 2009). The decline of the X-shaped structures and the accumulation of the linears are a direct readout of unhooking incisions. In XPF-ERCC1-depleted extract (Appendix Fig S1), the X-structures persist and the appearance of linear structures is greatly reduced (Fig 4B–D and Appendix Fig S1B–D), indicating the unhooking incisions are blocked (see also Klein Douwel *et al*, 2014). Addition of wild-type XPF-ERCC1 (XE$^{WT}$) rescues this incision defect (Fig 4B–D and Appendix Fig S1B–D) whereas the helicase-like domain mutants XE$^{L219R}$ or XE$^{C225R}$ did not (Fig 4B and Appendix Fig S1B). The XE$^{G314E}$ mutant did not inhibit incisions, while the XE$^{ΔNSGW}$ mutant caused a strong reduction in ICL unhooking (Fig 4C and Appendix Fig S1C). This is consistent with our earlier observation that the point mutant is functional in ICL repair while the deletion mutant is not. Finally, we found that the nuclease domain mutants XE$^{R670S}$ and XE$^{S767F}$ were unable to support efficient incisions (Fig 4D and Appendix Fig S1D). These results show that all our separation-of-function mutants are defective in ICL unhooking, which explains their inability to support ICL repair.

## The XPF$^{L219R}$-ERCC1 mutant complex is not recruited to the ICL

A possible explanation for why our mutants were not able to support ICL unhooking is that they are not recruited to the site of damage. To test this, we examined XPF recruitment by chromatin immunoprecipitation (ChIP). We replicated pICL in extract depleted of XPF-ERCC1 and supplemented with wild-type or mutant XPF-ERCC1 (Appendix Fig S2A), and performed chromatin immunoprecipitation with XPF antibodies at various time points. An unrelated plasmid (pQuant) was added to the reaction to determine background protein recruitment to undamaged DNA. The co-precipitated DNA was recovered and amplified by quantitative PCR with primers specific to the ICL region or to pQuant (Fig 5A). Using this assay, we recently showed that XPF is specifically recruited to the ICL at the time of unhooking incisions (Klein Douwel *et al*, 2014). The exact timing of recruitment can vary as a result of the immunodepletion procedure. We first examined the most N-terminal mutant XE$^{L219R}$ and found that, in contrast to the wild-type protein, recruitment of this mutant to the ICL was completely blocked (Fig 5B). In contrast, the XE$^{C225R}$ complex, containing a mutation just six residues further downstream, was recruited to the ICL as efficiently as the wild-type protein (Fig 5C and Appendix Fig S2B). We then examined the XE$^{ΔNSGW}$ mutant and found that it was recruited

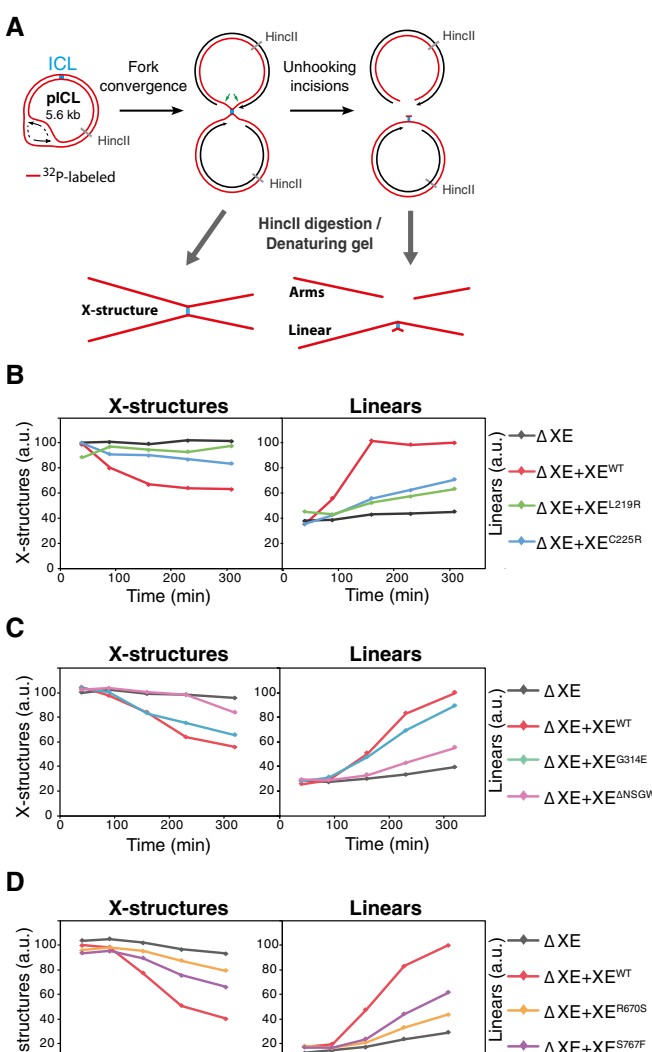

**Figure 4. XPF-ERCC1 separation-of-function mutants are all defective in ICL unhooking.**

A     Schematic representation of the assay used to directly measure unhooking incisions. $^{32}$P-labeled parental stands are indicated in red. Products before and after ICL unhooking during repair are indicated. HincII digestion of repair intermediates yields X-structures, arms, and linears under denaturing conditions.

B     XPF-ERCC1-depleted (ΔXE) or XPF-ERCC1-depleted egg extract complemented with wild-type (XE$^{WT}$) or mutant XPF-ERCC1 (XE$^{MUT}$) were incubated with pre-labeled pICL. Repair products were isolated at indicated times, linearized with HincII, separated on a denaturing agarose gel, and visualized by autoradiography. The X-structures and linear products were quantified and plotted.

C, D  As in (B) but using different XPF-ERCC1 mutant complexes. Experimental replicates are shown in Appendix Fig S1.

normally to the ICL (Fig 5D and Appendix Fig S2C). This is striking, because this region was shown to be important for the interaction between XPF and SLX4 (Andersen *et al*, 2009) and we have previously shown that SLX4 is important for the recruitment of XPF to the ICL. Lastly, we examined the recruitment of the nuclease domain mutants XE$^{R670S}$ and XE$^{S767F}$. Both mutants were recruited

to the ICL as efficiently as the wild-type protein (Fig 5E and F and Appendix Fig S6D).

We conclude that all mutants, except for XE$^{L219R}$, are recruited normally to the site of damage, suggesting the defect in incisions observed for these mutants is due to a defect in proper positioning of the nuclease within the repair complex.

### XPF leucine 219 is part of the major binding site between XPF and SLX4

To determine why XE$^{L219R}$ is not recruited to the ICL, we examined recruitment of both SLX4 and XPF to the ICL by ChIP. When we supplemented an XPF-ERCC1-depleted reaction with XPF-ERCC1 only, and not SLX4, XPF was not recruited to the ICL (Figs 6A and EV5A–C; and Klein Douwel et al, 2014). Supplemented SLX4 bound to the ICL and rescued the recruitment of wild-type XPF-ERCC1 (Figs 6A and EV5A–C), but not of the XE$^{L219R}$ mutant complex (Figs 6A and EV5A–C). These results show that a single point mutations can abrogate XPF recruitment and strongly suggest that this is caused by a defect in the direct interaction with SLX4.

To confirm this, we co-expressed FLAG-tagged XE$^{WT}$ and XE$^{L219R}$ with His-tagged SLX4 in Sf9 insect cells, immunoprecipitated XPF, and examined co-precipitation of SLX4. His-SLX4 was enriched after immunoprecipitation of wild-type XPF-ERCC1, but not XPF$^{L219R}$-ERCC1, indicating this mutant does not bind SLX4 (Figs 6B and EV5D). These findings indicate that XPF's leucine 219 is essential for the interaction between XPF and SLX4 and therefore required for the recruitment of XPF to the site of damage.

Two domains in SLX4 have been implicated in the interaction between SLX4 and XPF. The previously mentioned BTB domain and

the MUS312/MEI9 interaction-like, or MLR, domain (Fig 6C; Fekari et al, 2009; Kim et al, 2013). To further investigate the importance of the MLR domain for the interaction with XPF, we purified xlSLX4$^{WT}$ and xlSLX4$^{\Delta MLR}$. In contrast to wild-type SLX4, the ΔMLR mutant was not able to bind XPF from Xenopus egg extract (Fig 6D). This shows that the MLR domain of SLX4 acts as the major interaction site with XPF, which is in line with previous reports in human cells (Kim et al, 2013). Based on our data, this domain most likely interacts with leucine 219 of XPF.

Finally, we set out to examine the role of the interaction between the SLX4 BTB domain and XPF. The hsG325E mutation in XPF abrogates the interaction between XPF and the BTB domain of SLX4 (Andersen et al, 2009). We found that the XE$^{G314E}$ and XE$^{\Delta NSGW}$ mutants were both able to interact normally with full-length SLX4 (Figs 6B and EV5D). This is consistent with previous reports showing that the BTB domain is not essential for SLX4 and XPF interaction (Kim et al, 2013; Guervilly et al, 2015). One explanation for these observations is that this interaction is transient and can only be observed in the absence of the major interaction site involving the MLR domain. To be able to study this, we cloned and purified the SLX4 BTB domain alone and examined the interaction with XPF using size exclusion chromatography. However, even using an excess of BTB domain protein, we did not observe an interaction with XPF-ERCC1 (Fig EV5E). This indicates that this is not a high-affinity interaction site.

These findings, together with previous reports, support a model in which XPF and SLX4 interact through two binding sites (Fig 7). The first consists of the MLR domain of SLX4 and XPF leucine 219, and possibly a region around this residue. This is a high-affinity binding site that is responsible for the recruitment of XPF via SLX4

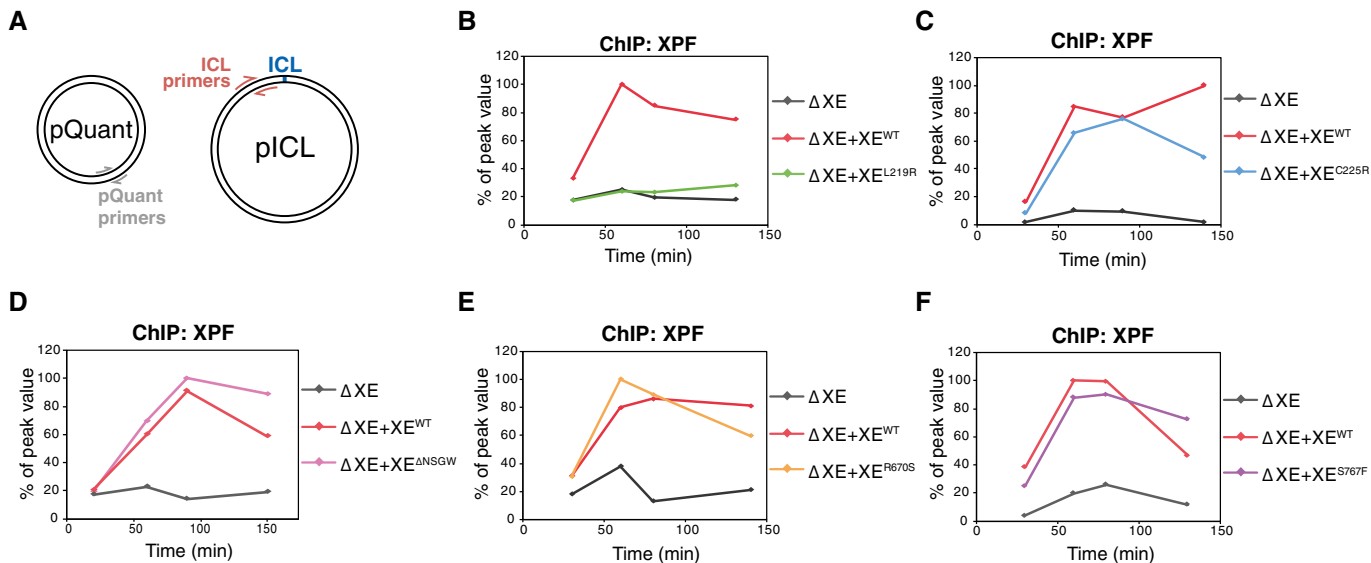

**Figure 5.  Recruitment of XPF-ERCC1 mutants to the ICL during repair.**

A    Schematic representation showing the primer locations on pICL and pQuant.

B    pICL was replicated in XPF-ERCC1-depleted (ΔXE) or XPF-ERCC1-depleted egg extract supplemented with wild-type (XE$^{WT}$) or mutant XPF-ERCC1 (XE$^{MUT}$; see Appendix Fig S2). Samples were taken at various times and immunoprecipitated with α-XPF antibodies. Co-precipitated DNA was isolated and analyzed by quantitative PCR using the primers depicted in (A). The qPCR data were plotted as the percentage of peak value with the highest value within one experiment set to 100%.

C–F    As in (B) but using the indicated XPF-ERCC1 mutant complexes. Experimental replicates are shown in Appendix Fig S2.

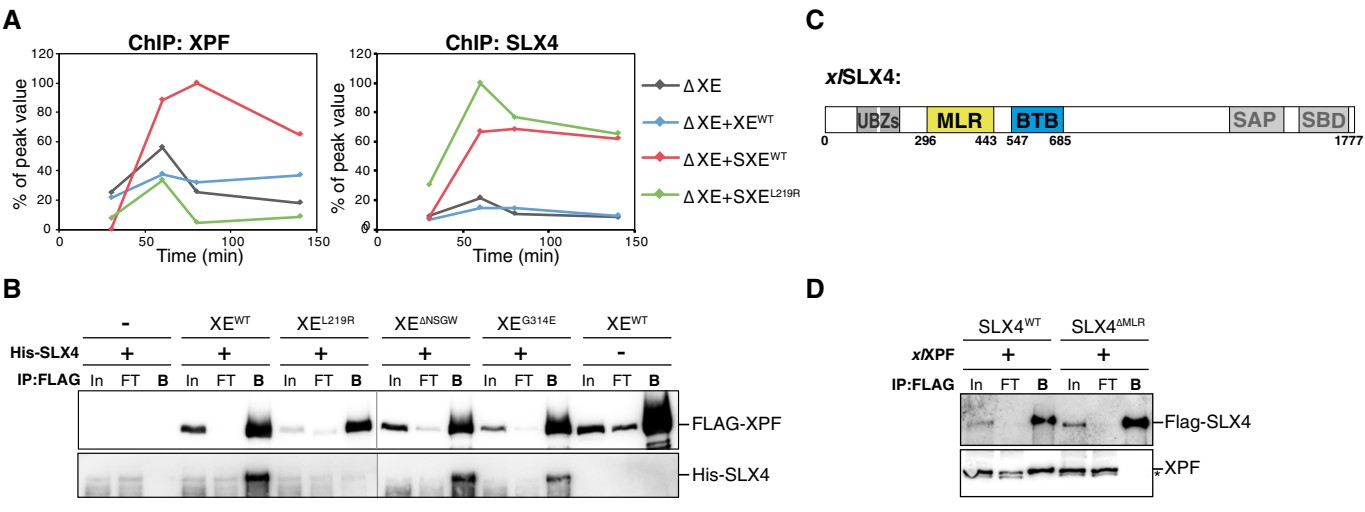

**Figure 6.  XPF leucine 219 is part of the major interaction site between XPF and SLX4.**

A  pICL was replicated in XPF-ERCC1-depleted (ΔXE) extract or in XPF-ERCC1-depleted extract supplemented with wild-type XPF-ERCC1 only (+XE^WT), wild-type XPF-ERCC1 and SLX4 (+SXE^WT), or XPF^L219R-ERCC1 and SLX4 (+SXE^L219R; see Fig EV5A). Samples were taken at the indicated times and immunoprecipitated with α-XPF (left panel) or α-SLX4 antibodies (right panel). Co-precipitated DNA was isolated and analyzed by quantitative PCR using ICL or pQuant primers. The qPCR data were plotted as the percentage of peak value with the highest value set to 100%.
B  Wild-type and mutant FLAG-XPF-ERCC1 were co-expressed with His-SLX4 in Sf9 insect cells. Cells were lysed and XPF was immunoprecipitated via the FLAG-tag. Samples were analyzed by Western blot using α-FLAG and α-His antibodies. In, input; FT, flow-through fraction; B, fraction bound to beads.
C  Schematic representation of *xl*SLX4 proteins, with the MLR and BTB domains indicated. Experimental replicates are shown in Fig EV5.
D  Purified wild-type FLAG-SLX4 and FLAG-SLX4^ΔMLR were added to *Xenopus* egg extract. SLX4 was immunoprecipitated via the FLAG-tag. Samples were analyzed by Western blot using α-FLAG and α-XPF antibodies. Line within blot indicates position where irrelevant lanes were removed. *, background band.

Source data are available online for this figure.

to ICLs. The second comprises the BTB domain of SLX4 and residues 312–315 of XPF. This interaction is transient, but important to promote nuclease activity of XPF possibly by orienting it properly.

# Discussion

Mutations in XPF-ERCC1 affect several DNA repair pathways and can cause multiple diseases likely due to differential inhibition of these pathways. Using *Xenopus* egg extracts, we have examined how certain mutations in XPF inhibit ICL repair, while maintaining proficient nucleotide excision repair. We have characterized five separation-of-function mutations that reside in the helicase-like and nuclease domains of XPF. While all these mutants are defective in ICL unhooking, this is caused by different mechanisms. The nuclease domain mutants are normally recruited to the ICL and most likely affect interactions with the DNA template or specific protein–protein interactions important for substrate recognition (Fig 7). The helicase-like domain mutants are part of a dual interaction site with SLX4. XPF's leucine 219 is part of a high-affinity interaction site that interacts with the MLR domain of SLX4, while deletion of residues 312–315 of XPF disrupts a transient second interaction site with the BTB domain of SLX4 (Fig 7 and Table 1).

We have previously shown that XPF-ERCC1 is recruited to the site of damage by SLX4 (Klein Douwel *et al*, 2014). A specific residue in the helicase-like domain of XPF has been implicated in the interaction with SLX4 (Yildiz *et al*, 2002; Andersen *et al*, 2009). A glycine to glutamic acid mutation at residue 325 of human XPF abrogated the interaction with SLX4 in a yeast two-hybrid assay

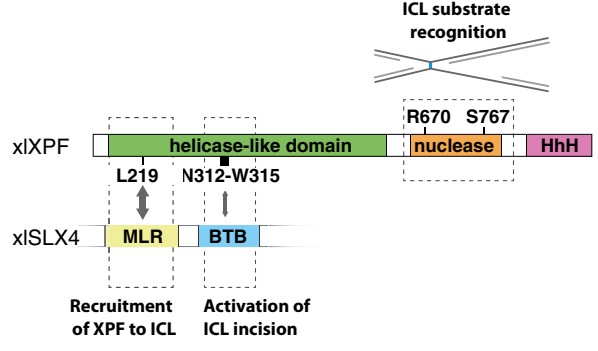

**Figure 7.  Model for ICL repair-specific features of XPF.**
Leucine 219 in the helicase-like domain of XPF is essential for the interaction of XPF with the MLR domain of SLX4. This interaction mediates the recruitment of XPF to an ICL. Residues 312–315 transiently interact with the BTB domain of SLX4 and are required for the incisions of an ICL by XPF. Arginine 670 and serine 767 in the nuclease domain of XPF are crucial for the recognition of the ICL substrate.

(Andersen *et al*, 2009). This yeast two-hybrid assay was performed with C-terminal deletion mutants of SLX4 and the interaction with XPF was pinpointed to the BTB domain. However, these mutants lacked the MLR domain which has also been implicated in the interaction with XPF (Kim *et al*, 2013). In our hands, the equivalent mutation in *xl*XPF, G314E, did not abrogate ICL repair or recruitment. Moreover, a deletion mutant in which this glycine and three additional residues around it were removed (XPF^ΔNSGW) interacted

**Table 1. Summary of features of different XPF mutations.**

| xl | hs | Patient mutation | Clinical features | Nuclease activity | DNA binding | NER | ICL repair | Unhooking | Recruitment to ICL | SLX4 binding |
|---|---|---|---|---|---|---|---|---|---|---|
| XPF$^{L219R}$ | XPF$^{L230R}$ | L230P | FA | + | + | + | − | − | − | − |
| XPF$^{C225R}$ | XPF$^{C236R}$ | C236R | CS v CS/XP/FA | + | + | + | − | − | + | N.D |
| XPF$^{G314E}$ | XPF$^{G325E}$ | N.A. | N.A. | + | + | + | + | + | N.D. | N.D. |
| XPF$^{\Delta NSGW}$ | XPF$^{\Delta NSGW}$ | N.A. | N.A. | + | + | + | − | − | + | + |
| XPF$^{R670S}$ | XPF$^{R689S}$ | R689S | FA | − | + | + | − | − | + | N.D. |
| XPF$^{S767F}$ | XPF$^{S786F}$ | N.A. | N.A. | + | + | + | − | − | + | N.D. |

Abbreviations are as follows: +, normal; − absent or defective; NA, not applicable; ND, not determined.

normally with SLX4. Based on these observations, we suggest that the interaction site between the BTB domain of SLX4 and residues 312–315 of XPF is a minor interaction site. This is consistent with our data showing that the isolated BTB domain does not interact strongly with XPF and with results reported by Guervilly *et al* that show only a slight decrease in XPF binding after mutation of the SLX4 BTB domain (Guervilly *et al*, 2015). However, this interaction is important because the XPF$^{\Delta NSGW}$-ERCC1 mutant is deficient in ICL unhooking and repair (Figs 2 and 4). Therefore, we propose that this transient interaction site is important for activation of XPF-ERCC1 by ensuring correct positioning onto its substrate. Interestingly, SLX4 has been shown to stimulate XPF-ERCC1 activity on model substrates (Hodskinson *et al*, 2014) which could be mediated through this interaction. Although we cannot exclude that this ICL repair defect of the XPF$^{\Delta NSGW}$-ERCC1 mutant is caused by a different mechanism, the fact that this mutant completely overlaps with a previously identified interaction site strongly supports this explanation.

We and others have shown that the MLR domain of SLX4 is essential for the interaction with XPF (Fig 6 and Kim *et al*, 2013), but it was not known which site on XPF was involved in this interaction. We now show that the XPF leucine 219 to arginine mutant is defective in binding to SLX4 indicating this is the site that interacts with the MLR domain. Further examination of the residues surrounding leucine 219 is required to better characterize this interaction site. Notably, our ChIP results indicate that the cysteine at position 225 is not required for the interaction. We further show that the XPF$^{L219R}$-ERCC1 mutant complex is defective in ICL repair but not in NER. This shows that the interaction with SLX4 is specific to the role of XPF-ERCC1 in ICL repair. This is consistent with the fact that patients with an L230P mutation suffer from FA and not XP. While previously it was assumed that poor stability of the XE$^{L230P}$ protein was causing the FA phenotype, these data suggest that a functional defect, namely impaired interaction with SLX4, may cause, or contribute to, the disease.

Interestingly, the XPF$^{C225R}$-ERCC1 mutant does affect ICL recruitment but is still defective in ICL repair. Possibly this mutant does not prevent binding to SLX4, but does affect the interaction site in a way that it cannot properly position XPF for incisions. Two patients have been identified carrying the C236R (*xl*C225R) mutation, both show Cockayne syndrome (CS) phenotypes, while only one patient additionally shows a Fanconi anemia phenotype. The Cockayne syndrome phenotype is thought to be caused by a specific defect in transcription-coupled nucleotide excision repair (TC-NER). While

mutations in XPF are not expected to specifically affect TC-NER, because it acts downstream in the NER pathway where the transcription-coupled and global NER pathways have come together, a CS phenotype has been observed previously in patients with mutations in XPF (Kashiyama *et al*, 2013). Our data show that the *xl*C225R mutation prevents ICL repair, but does not affect NER. However, *Xenopus* egg extracts are transcription incompetent and we may therefore not identify a defect in TC-NER. Why only one of the patients carrying the C236R mutation presents with a clear FA phenotype is currently unclear. Possibly the other patient has an additional mutation or specific genetic background that neutralizes the ICL repair defect.

In addition to the helicase-like domain mutants, we found two separation-of-function mutants in the nuclease domain of XPF. Arginine 670 is located within the active site of *xl*XPF and mutating it to a serine severely reduces nuclease activity as was shown by us and others (Bogliolo *et al*, 2013). Nevertheless, our data indicate that the XE$^{R670S}$ mutant can still support NER to wild-type levels. This is in line with previous data showing the human equivalent, *hs*XE$^{R689A}$, can incise an NER substrate (Enzlin & Schärer, 2002; Staresincic *et al*, 2009; Su *et al*, 2012). Interestingly, the human mutant protein did show a shift in incision position, suggesting the residue is not directly involved in catalysis, but contributes to the proper orientation of the active site onto the DNA substrate (Su *et al*, 2012). This aberrant positioning is apparently not detrimental for NER but does prevent ICL repair in our assays. This is a likely explanation as the DNA template for incision differs in both repair pathways. Moreover, it is supported by the identification of a patient with the *hs*R689S mutation that suffers from FA, but not XP (Bogliolo *et al*, 2013).

The S767F mutation is also located in the nuclease domain and structure predictions based on the crystal structure of the nuclease domain of archaeal XPF in complex with DNA, indicating that it could be involved in protein–DNA interaction (Newman *et al*, 2005). In our experiments, this mutant shows a mild reduction in nuclease activity, is proficient in NER but largely deficient in ICL repair. We propose that, like the arginine 670, this residue is important in positioning the active site specifically on an ICL template likely by direct contact with the DNA. We did not observe reduced DNA binding affinity for these mutants most likely because XPF-ERCC1 contains multiple DNA interacting domains and it was shown that mutation of at least two of those is required to reduce this affinity (Su *et al*, 2012).

XPF-ERCC1 is essential for the repair of ICLs induced by chemotherapy agents, such as derivatives of cisplatin and nitrogen

mustards (Kirschner & Melton, 2010). Moreover, high expression of ERCC1 has been associated with poor response to chemotherapy in many cancers and could be a potential target to overcome resistance (McNeil & Melton, 2012). A better understanding of the ICL repair function of XPF-ERCC1 could potentially lead to the design of ICL-specific inhibitors that could be beneficial in cancer treatment.

# Materials and Methods

### Protein expression and purification

His-tagged *hs*ERCC1 was cloned into pDONR201 (Life Technologies). FLAG-tagged *xl*XPF was cloned into pFastBac1 (Life Technologies) and in pDONR201. The XPF mutations (L219R, C225R, G314E, ΔNSGW, R670S, D668A, and S767F) and ERCC1 mutation (K247A/K281A) were introduced in pDONR-XPF using QuikChange site-directed mutagenesis protocol. Baculoviruses were produced using the BAC-to-BAC system (*xl*XPF$^{WT}$), or the BaculoDirect system (*hs*ERCC1 and *xl*XPF$^{MUTs}$) following manufacturer's protocol (Life Technologies). Proteins were expressed in suspension cultures of Sf9 insect cells by co-infection with His-*hs*ERCC1 (or His-*hs*ERCC1$^{K247A/K281A}$) and FLAG-*xl*XPF (or FLAG-*xl*XPF mutants) viruses for 72 h. Cells from 750 ml culture were collected by centrifugation, resuspended in 30 ml of lysis buffer (50 mM K$_2$HPO$_4$ pH 8.0, 500 mM NaCl, 0.1% NP-40, 10% glycerol, 0.4 mM PMSF, 1 tablet/50 ml Complete Mini EDTA-free Protease Inhibitor Cocktail (Roche), 10 mM imidazole), and lysed by sonication. The soluble fraction obtained after centrifugation (40,000 × *g* for 40 min at 4°C) was incubated for 1 h at 4°C with 1 ml of Ni-NTA-agarose (Qiagen) that was pre-washed with lysis buffer. After incubation, the beads were washed using 50 ml of wash buffer (50 mM K$_2$HPO$_4$ pH 8.0, 300 mM NaCl, 0.1% NP-40, 10% glycerol, 0.1 mM PMSF, 10 μg/ml apropotin/leupeptin, 20 mM imidazole). The *xl*XPF-*hs*ERCC1 complex was eluted in elution buffer (50 mM K$_2$HPO$_4$ pH 8.0, 300 mM NaCl, 0.1% NP-40, 10% glycerol, 0.1 mM PMSF, 10 μg/ml apropotin/leupeptin, 250 mM imidazole). The eluate was diluted with FLAG-wash buffer I (20 mM K$_2$HPO$_4$ pH 8.0, 200 mM NaCl, 0.1% NP-40, 10% glycerol, 0.4 mM PMSF) and incubated for 1 h at 4°C with 500 μl of anti-FLAG M2 affinity gel (Sigma) that was pre-washed with FLAG-wash buffer I. After incubation, the beads were washed with 30 ml of FLAG-wash buffer I, and subsequently with 30 ml of GF buffer (25 mM Hepes pH 8.0, 200 mM NaCl, 10% glycerol, 5 mM β-mercaptoethanol). The *xl*XPF-*hs*ERCC1 complex was eluted in 3 ml of GF buffer containing 100 μg/ml 3× FLAG peptide (Sigma). The protein was then loaded onto a HiLoad 16/600 Superdex 200 pg gel filtration column (GE Healthcare) equilibrated in GF buffer. Fractions containing the *xl*XPF-*hs*ERCC1 heterodimer were eluted between 60 and 70 ml (Enzlin & Schärer, 2002), pooled, and concentrated with an Amicon Ultra-4 centrifuge filter unit, 30 kDa (Merck Millipore). Protein was aliquoted, flash-frozen, and stored at −80°C. FLAG-tagged *xl*SLX4 was purified as previously described (Klein Douwel *et al*, 2014). His-tagged *xl*SLX4 was cloned into pDONR201 (Life Technologies) and baculoviruses were produced using the BaculoDirect system following manufacturer's protocol (Life Technologies). His-tagged *xl*SLX4 was expressed in 150 ml suspension cultures of Sf9 insect cells for 72 h. Cells were collected by centrifugation, resuspended in lysis buffer (50 mM Tris pH 8.0,

500 mM NaCl, 0.1% NP-40, 10% glycerol, 0.4 mM PMSF, 10 mM imidazole, 1 tablet/10 ml Complete Mini EDTA-free Protease Inhibitor Cocktail (Roche)), and lysed by sonication. The soluble fraction obtained after centrifugation (40,000 × *g* for 40 min at 4°C) was incubated for 1 h at 4°C with 750 μl of Ni-NTA-agarose (Qiagen) that were pre-washed with lysis buffer. After incubation, the beads were washed with wash buffer (50 mM Tris pH 8.0, 300 mM NaCl, 0.1% NP-40, 5% glycerol, 0.1 mM PMSF, 20 mM imidazole, 10 μg/ml apropotin/leupeptin). His-tagged *xl*SLX4 protein was eluted in wash buffer containing 250 mM imidazole. The protein was aliquoted and stored at −80°C. The pDONR201 construct for FLAG-tagged *xl*SLX4 was used to create the *xl*SLX4$^{ΔMLR}$ mutant. PCR amplification was used to replace the MLR domain with a short linker containing a KpnI restriction site. Expression and purification of FLAG-tagged *xl*SLX4$^{ΔMLR}$ were identical to the wild-type protein.

### Nuclease assay

Nuclease assay was performed as previously described (De Laat *et al*, 1998). The following primers were obtained (Integrated DNA technologies): SL: 5′-FAM-CGCCAG CGC TCGGTTTTTTTTTTTTTT TTTTTTTTTCCGAGCGCTGGC-'3; F1: 5′-FAM- CGCGATGCGG ATCC AA-3′; F2: 5′-CCTAGACTTAAGAGGCCAGACTTGGATCCGCATCGC-3′; F3: 5′-GGCCTCTTAAGTCTAGG-3′. For the stem-loop structure, primer SL was heated for 3 min at 95°C, followed by stepwise cooling to allow annealing (30 min at 60°C, 30 min at 37°C, 30 min at 25°C, 30 min on ice). To assemble the 3′ flap substrate, primer F1 was annealed to primer F2 and F3 in a 1:1:1 ratio and annealed similar to the stem-loop substrate. Nuclease reactions (15 μl) were carried out in nuclease buffer (50 mM Tris pH 8.0, 0.2 mM MnCl$_2$, 0.1 mg/ml bovine serum albumin, and 0.5 mM β-mercaptoethanol) containing 100 nmol of substrate DNA and 10–100 nM of recombinant XPF-ERCC1 wild-type or mutant protein complex. Reactions were incubated for 30 min at room temperature and stopped by addition of 15 μl denaturing PAGE Gel Loading Buffer II (Life Technologies, Inc.). Samples were heated to 72°C for 3 min, snap-cooled, and loaded onto a 12% denaturing urea–PAGE gel. Gels were directly measured on a Typhoon phosphor imager (GE Healthcare) on the blue 488 channel.

### Fluorescent anisotropy binding assay

Increasing concentrations of protein were incubated with 10 nM of a 3′ flap DNA substrate containing a 5′-fluorescent FAM label (see Nuclease assay). The reaction was incubated with annealing buffer (25 mM Hepes pH 8.0, 15% glycerol, 0.1 mg/ml BSA, 2 mM CaCl$_2$) in a 384-well plate (kBioscience) for 1 h at room temperature, and fluorescent anisotropy was measured on a Spectramax I3 (Molecular Devices). The data were fitted using Origin 8.5 to the equation $y = a + b × x^n/(k^n + x^n)$, where $x$ is the protein concentration, $y$ is the fluorescence anisotropy, and $k$ is the $K_d$ value.

### DNA replication and repair assay in *Xenopus* egg extracts

DNA replication and preparation of *Xenopus* egg extracts (HSS and NPE) were performed as described previously (Walter *et al*, 1998; Tutter & Walter, 2006). Preparation of plasmid with a site-specific cisplatin ICL (pICL), and ICL repair assays were performed as

described (Räschle *et al*, 2008; Enoiu *et al*, 2012). Briefly pICL was incubated with HSS for 20 min, following addition of two volumes of NPE (*t* = 0) containing $^{32}$P-α-dCTP. Aliquots of replication reaction (4–10 μl) were stopped at various times with ten volumes of Stop Solution II (0.5% SDS, 10 mM EDTA, 50 mM Tris pH 7.5). Samples were incubated with RNase (0.13 μg/μl) followed by proteinase K (0.5 μg/μl) for 30 min at 37°C each. DNA was extracted using phenol/chloroform, ethanol-precipitated in the presence of glycogen (30 mg/ml), and resuspended in 5–10 μl of 10 mM Tris pH 7.5. ICL repair was analyzed by digesting 1 μl of extracted DNA with HincII, or HincII and SapI, separation on a 0.8% native agarose gel, and quantification using autoradiography. Repair efficiency was calculated as described (Knipscheer *et al*, 2012).

## Unscheduled DNA synthesis

The assay to monitor unscheduled DNA synthesis (UDS) in *Xenopus* egg extract was adapted from Gaillard *et al* (1996). A 6.25-μl reaction containing 2.5 μl HSS and 6 ng/μl non-treated or UV-C-irradiated (350 μJ/m$^2$) pControl was supplemented with 5 mM MgCl$_2$, 0.5 mM DTT, 4 mM ATP, 40 mM phosphocreatine, 0.5 μg creatine phosphokinase, and 80 μCi/ml $^{32}$P-α-dCTP (3,000 Ci/mmol). Reactions were incubated at room temperature for 2 h and stopped by addition of ten volumes of Stop Solution II (0.5% SDS, 10 mM EDTA, 50 mM Tris pH 7.5). Samples were incubated with proteinase K (0.5 μg/μl) for 30 min at 37°C. DNA was extracted using phenol/chloroform, ethanol-precipitated in the presence of glycogen (30 mg/ml), and resuspended in 5–10 μl of 10 mM Tris pH 7.5. Extracted DNA (2 μl) was digested with HincII and separated on a 0.8% native agarose gel. The gel was stained with SYBR GOLD (Fisher) and subsequently dried and quantified using autoradiography. The background signal from non-treated plasmid was subtracted. To compare the levels of UDS between experiments, the signal for wild-type XPF-ERCC1 add back was set to 100%. This is because the efficiency of UDS can differ between extract preparations and depletions.

## Antibodies and immunodepletions

Antibodies were raised against residues 444–797 of *xl*XPF, full-length *xl*ERCC1, and residues 825–1,052 of *xl*SLX4. Specificity was confirmed using Western blot (Klein Douwel *et al*, 2014). XPF-ERCC1 was removed from extract using three rounds of depletion with the α-ERCC1 serum (HSS and NPE). ERCC1 depletion was described previously (Klein Douwel *et al*, 2014). For the unscheduled DNA synthesis assay, HSS was depleted using three rounds of an ERCC1 depletion (one volume of PAS was bound to one volumes of anti-serum or pre-immune serum, and added to four volumes of HSS) followed by three rounds of depletion with the α-XPF serum (one volume of PAS was bound to three volumes of anti-serum or pre-immune serum, and added to five volumes of HSS). Anti-FLAG M2 antibody was purchased from Sigma and His-antibody from Westburg.

## Incision assay

Incision assay was performed as described in Klein Douwel *et al* (2014). Briefly, pICL and pQuant were labeled via nick translation.

pQuant was added as an internal control to allow accurate calculation of incision efficiency. pICL (225 ng) and pQuant (11.25 ng) were incubated in 1.5 units of NB-BSR DI enzyme (NEB) and 1× NEBuffer 2 for 30 min at room temperature. Subsequently, 11 μl of DNA Polymerase I mix (5 units of DNA polymerase I (NEB), dATP, dGTP, dTTP (0.5 mM each), dCTP (0.4 μM), $^{32}$P-α-dCTP (3.3 μM) in 1× NEBuffer 2) was added and this was incubated for 3 min at 16°C. The reaction was stopped with 180 μl of Stop Solution II, treated with proteinase K, and phenol/chloroform-extracted. Excess label was removed using a Micro Bio-Spin 6 Column (Bio-Rad). After ethanol precipitation, the pellet was resuspended in 5 μl of ELB (10 mM HEPES-KOH pH 7.7, 50 mM KCl, 2.5 mM MgCl$_2$, and 250 mM sucrose). The labeled plasmid (pICL*) was used in a replication reaction and samples at various times were extracted and digested with HincII. Fragments were separated on a 0.8% alkaline denaturing agarose gel for 18 h at 0.85 Volts/cm, after which the gel was dried and exposed to a phosphor screen. Quantification was performed using ImageQuant software (GE healthcare). The highest value was set at 100% for the X-shape and the linear products.

## Chromatin immunoprecipitation

Chromatin immunoprecipitation (ChIP) was performed as described (Pacek *et al*, 2006). Briefly, reaction samples were crosslinked with formaldehyde, sonicated to yield DNA fragments of roughly 100–500 bp, and immunoprecipitated with the indicated antibodies. Protein–DNA crosslinks were reversed and DNA was phenol/chloroform-extracted for analysis by quantitative real-time PCR with the following primers: ICL (5′-AGCCAGATTTTTCCTCCTCTC-3′ and 5′-CATGCATTGGTTCTGCACTT-3′) and pQuant (5′-TACAAATGTACGGCCAGCAA-3′ and 5′-GAGTATGAGGGAAGCGGTGA-3′). The values from pQuant primers were subtracted from the values for pICL primers.

## Immunoprecipitations

Proteins were expressed in adherent cultures of Sf9 insect cells in 6-well plates by co-infection with His-*hs*ERCC1, His-*xl*SLX4, and FLAG-*xl*XPF (or FLAG-*xl*XP mutants) viruses for 72 h. Cells were resuspended in medium and collected by centrifugation, resuspended in 250 μl of lysis buffer (50 mM Tris pH 8.0, 300 mM NaCl, 1% Triton, 4 mM EDTA, 10 μg/ml apropotin/leupeptin), and lysed by sonication. After centrifugation (20,000 × *g* for 20 min at 4°C), 200 μl of soluble fraction was incubated for 30 min at 4°C with 8 μl of FLAG M2 beads (Sigma-Aldrich) that were pre-washed with lysis buffer. After incubation, the beads were washed using 2 ml lysis buffer. Beads were taken up in 50 μl of 2× SDS sample buffer and incubated for 5 min at 95°C. Proteins were loaded on SDS–PAGE and visualized by Western blot using respective antibodies.

For immunoprecipitations from *Xenopus* egg extract, FLAG-tagged *xl*SLX4$^{WT}$ or *xl*SLX4$^{ΔMLR}$ protein was added to NPE/HSS at a concentration of 5 ng/μl. To each 20 μl extract, 45.5 μl of IP buffer (1×ELB salts, 0.25 M sucrose, 75 mM NaCl, 2 mM EDTA, 10 μg/ml apropotin/leupeptin, 0.1% NP-40) and 10 μl pre-washed FLAG M2 beads (Sigma-Aldrich) were added. Beads were incubated for 90 min at 4°C and subsequently washed using 2.5 ml IP buffer. Beads were taken up in 30 μl of 2× SDS sample buffer and incubated

for 5 min at 95°C. Proteins were loaded on SDS–PAGE and visualized by Western blot using respective antibodies.

Expanded View for this article is available online.

## Acknowledgements

This work was supported by the Netherlands organization for Scientific Research (VIDI 700.10.421 to P.K.) and a project grant from the Dutch Cancer Society (KWF HUBR 2015-7736 to P.K.). We thank N. Martin-Pintado for assistance with the anisotropy assays and suggestions on the manuscript. We also thank J.C. Walter for the *xl*PCNA antibody and K.A. Cimprich for the *xl*XPA antibody. In addition, we thank J.C. Walter and J.A.F. Marteijn for suggestions on the manuscript, the Hubrecht animal caretakers for animal support, and the other members of the Knipscheer laboratory for feedback.

## Author contributions

DKD and PK designed the experiments. DKD performed the majority of the experiments. WSH performed IP assays from extract and insect cells. RACMB performed some of the ChIP assays. DKD analyzed the data, and DKD and PK discussed the data with the help of the other authors. DKD and PK wrote the manuscript and WSH and RACMB commented on the manuscript.

## Conflict of interest

The authors declare that they have no conflict of interest.

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
