## [Review Process File · The EMBO Journal]

Manuscript EMBO-2016-95223

Recruitment and positioning determine the specific role of the XPF-ERCC1 endonuclease in interstrand crosslink repair

Daisy Klein Douwel, Wouter Hoogenboom, Rick Boonen and Puck Knipscheer

Corresponding author: Puck Knipscheer, Hubrecht Institute

Review timeline:

Submission date:	11 July 2016
Editorial Decision:	06 September 2016
Revision received:	27 December 2016
Editorial Decision:	23 January 2017
Revision received:	07 February 2017
Accepted:	14 February 2017

Editor: Hartmut Vodermaier

Transaction Report:

1st Editorial Decision

06 September 2016

Thank you again for submitting your manuscript on the characterization of XPF-ERCC1 separation-of-function mutants to The EMBO Journal. I would like to apologize for the delay in its evaluation, which was due to limited referee availability and the need for extended review times during the summer vacation season. We have now received the complete set of referee reports on your study, which I am enclosing below for your information.

As you will see, all referees acknowledge the technical quality and potential interest of the presented data. However, referee 2 raise some concerns regarding the overall advance conveyed by your new results, and referee 4 criticizes aspects of analysis and presentation in both figures and text.

Faced with these mixed recommendations, I would like to give you an opportunity to address the referees' concerns via a revised manuscript. In this respect, I realize that novelty concerns may be in parts already alleviated by a crisper and more concise presentation and focus on the particularly novel insights and implications; however, I feel it will also be important to extend the mutant characterization to obtain some deeper understanding into the molecular basis of ICL/NER functional separation of some of the mutants, as requested by referee 2 (and to some extent also by the other two reviewers).

Should you be able to adequately improve these two key aspects, we would be happy to consider this study further for publication. However, please remember that it is our policy to allow only a single round of major revision, making it important to carefully respond to all points raised during this round. As usual, any related/competing work published during the revision period will have no negative impact on our final assessment of your revised study. Further information regarding

submission of revised manuscript can be found below and in our Guide to Authors.

REFEREE REPORTS

Referee #2:

Knipscheer and colleagues present an interesting study, which reports isolation of separation of function Xpf mutations. These mutations affect repair of ICL repair without affecting repair of UV damage. The mutations were chosen for analysis were from taken reports in the literature of Xpf mutations in different disease syndromes. Two were designed based on reports of mutations that disrupt interaction with Slx4. The equivalent mutations were made in *Xenopus* Xpf and introduced into *Xe* extracts depleted of Xpf-Ercc1-Slx4 and rescue experiments to analyse DNA repair were carried out. The bottom line is isolation of three mutations that are deficient in the unhooking step of ICL repair but which don't affect ICL repair. One of the mutations blocks XPF recruitment by blocking the binding of XPF to SLX4.

The data in this paper are interesting and of a high technical quality. The main concern in terms of suitability specifically for EMBO is that the conclusions are not particularly surprising or novel. For example, it has already been demonstrated that the ICL repair role of Xpf can be separated genetically from the NER role. This isn't a novel finding. Also, cells deficient in Xpf or Slx4 show ICL repair defects; Xpf-deficient cells are NER-defective but Slx4-deficient cells are not. So mutations in Xpf that prevent binding to Slx4 would be expected to inhibit ICL repair without affecting NER. This is not surprising or unanticipated. Similarly, the Knipscheer lab has already shown the role of Xpf in ICL repair is at the unhooking stage, so it's not surprising that the ICL repair-deficient mutants in Xpf are deficient in unhooking. That's the step Xpf catalyzes. What would be novel would be some insight into why it is that the ICL repair-defective mutants do not affect NER. Getting some handle on the difference between Xpf in the two different contexts would represent an important advance. But this paper doesn't go there, and so in its current form its suitable for a more specialist journal (JBC?).

One other point: two mutants which do not bind to Slx4 are used - G314E and deltaNGWS. The former is proficient for ICL repair while the latter is not. The explanation given is that G314E doesn't disrupt the Slx4 interaction sufficiently to inhibit ICL repair. But there's no data on the difference between the two mutants in terms of Slx4 binding - they may be equally defective. If so, it's possible that the NGWS affects an aspect of Xpf function other than Slx4 interaction which explains the ICL repair defect. It's true that the L219R mutation inhibits Slx4 interaction and ICL repair but it's simply correlation. Having an Xpf mutant that doesn't interact with Slx4 and that doesn't affect ICL repair argues the interaction with Slx4 is not important for ICL repair and this would need to be addressed.

Referee #3:

In a previous paper in *Mol Cell* in 2014, the authors showed that ERCC1-XPF is responsible for the unhooking step in interstrand crosslink (ICL) repair and that this activity depends on an interaction with SLX4. In the present paper, they analyze how a number of mutant alleles of XPF, isolated from FA patients and in screens in *Drosophila* affect ICL repair and nucleotide excision repair (NER - XPF was first described as an NER protein).

The authors use their *Xenopus* oocyte system to monitor replication-dependent ICL repair and expressed and purified six mutant XPF proteins. Consistent with the patient and cellular phenotypes of these proteins, they are proficient in NER, while displaying a defect in ICL repair. Importantly, the authors can pinpoint the ICL repair defect to the unhooking step, answering a long-standing question in the field, whether the key role of XPF in ICL repair is in the unhooking or a later step in recombination. Interestingly, with the exception of the L219P mutant all of the proteins are recruited to sites of ICL repair and interact with SLX4, showing that their defect in ICL repair results from a

defect in positioning at the incision site, either through a defect in the interaction with DNA around the nuclease active site (R670S, S767F), or a defect due positioning in the ICL repair complex likely by protein-protein interaction (C225R, ΔNSGW). By contrast L219P does not bind SLX4 and is not recruited to ICL repair sites.

This work is carried out to a high technical standard and the great care that was taken to ensure that none of the defects observed was due to protein aggregation, a common problem with XPF mutant proteins, is particularly impressive. The work significantly adds to our understanding of the mechanisms by which XPF contributes to ICL repair.

A few minor concerns should be addressed prior to publication:

- 1) p3. line 16: It would be worth mentioning that all XP patients with residual NER activity have a relatively mild phenotype, due to residual levels of XPF protein. This low levels of XPF proteins are believed to sufficient to support ICL repair as the patients do not suffer from FA (Ahmad et al, PLoS Genet, 2010, 6, e1000871)
- 2) p4. bottom. The ability of ERCC1-XPF to incise both sides of an ICL was first shown by Wood and coworkers (Kuraoka et al, JBC, 2000, 275, 26632). This reference should be cited here.
- 3) p6. line 5: Ahmad et al, PLoS Genet, 2010, 6, e1000871 should be cited as the reference characterizing the cellular status of XPF in XP-F patients.
- 4) P12. bottom: Bergstralh and Sekelsky, Trends Genet 2008, 24, 70 would be good to cite for a discussion of how XPF might function downstream of incision in ICL repair.
- 5) P14. line 10 - An alternative explanation for why XPF-ΔNSGW is defective in ICL repair is that it may well have an interaction defect with SLX4, but that this defect does not affect recruitment to sites of ICLs, but rather the position of the two proteins at the sites of ICL incision site. It is entirely possible that there are multiple sites of interaction between SLX4 and XPF. This point should be considered here, on p.15 (line 10) and in the discussion on p.16/17.

Referee #4:

In this paper Knipscheer explores the role of the excision nuclease Xpf-Ercc1 in replication coupled DNA crosslink repair. Building on her own pioneering work with J Walter where together they established how a DNA crosslink is repaired during DNA replication using a *Xenopus* egg extract cell free system. In later follow up work Knipscheer established how the key excision or unhooking step occurs through the interplay between the FA pathway and Slx4 with Xpf-Ercc1 (Molecular Cell 2014). Now in this paper they carry out a more detailed analysis of the nuclease complex with Slx4, more specifically they exploit single mutations in the Xpf/Ercc1 complex that co-segregate with a human syndrome with overlap with FA and Cockayne syndrome. These mutations cause DNA crosslinker sensitivity but appear not to impact on the role of this complex in nucleotide excision repair.

The main conclusions of this current study is that mutants of Xpf-Ercc1 complex can be divided into two groups - those that impact both NER and CX repair (nuclease active site mutants), those that work in NER but do not unhook DNA crosslinks. This latter group falls into two further groups - those that bind Slx4 and are then not recruited to the crosslink, and those that bind Slx4 are recruited to the Cx but fail to unhook it. They correctly conclude from this work that Slx4/ Xpf-Ercc1 complex not only works to recruit the nuclease to the damage site but also must somehow position the nuclease once it is there so that it cuts this substrate.

These are novel and important insights into DNA CX repair and therefore merits very strong consideration for publication in EMBO. However despite my obvious enthusiasm for the work I would like the authors to address my following concerns.

1. Generally the manuscript is written rather poorly, the introduction is way too long and utterly boring to read. After several espresso cups I did manage to wade through this tome akin to reading War and Peace in one setting for a romantically disinclined individual. It should be 1/3 the size and can be much more succinct.
2. Generally the figures are just appallingly laid out. I use the test that one should be able to "read" the figures without having to look at the text. I found this impossible in this case. The figures in her and Walters papers are really excellent - she should consider emulating these or at the very least matching their quality and clarity.
3. The last figure should have a model that sums up what they are saying in the paper. Without such a visual encapsulation of the message its impact will be lost to many.
4. Fig 1 is excellent then all this goes rather downhill. Fig 2 WB should have a loading control. Perhaps space should be devoted to the Walter Knipsheer model and how this is assayed in their graphs.
5. Figure 3 : Awful ! Have they shown that depleting other components of NER abrogates UDS in this assay ? 2B why is the UDS analysis only provided for one of the mutants and not all of them shown in the panel below. 2C there is no stats here and I am concerned about the range of intact excision repair, for instance some the mutants are even better than the wild type (the last one for instance). I would also like to see this data corroborated by showing by slot blot that CPD dimers are removed (there are good antibodies against these lesions that track their removal. Getting this figure right matters since the rest of the paper builds on the mutants that work in NER but not in CX repair.
6. Figure 4 again difficult to follow- work in improving clarity here . Excessively cropped WB are not really acceptable these days, also again no loading control.
7. Figure 6 no loading control.

1st Revision - authors' response

27 December 2016

We thank the reviewers for their feedback on our work. Our response to their comments is shown in blue italics.

Referee #2:

Knipscheer and colleagues present an interesting study, which reports isolation of separation of function Xpf mutations. These mutations affect repair of ICL repair without affecting repair of UV damage. The mutations were chosen for analysis were from taken reports in the literature of Xpf mutations in different disease syndromes. Two were designed based on reports of mutations that disrupt interaction with Slx4. The equivalent mutations were made in *Xenopus* Xpf and introduced into *Xe* extracts depleted of Xpf-Erc1-Slx4 and rescue experiments to analyse DNA repair were carried out. The bottom line is isolation of three mutations that are deficient in the unhooking step of ICL repair but which don't affect ICL repair. One of the mutations blocks XPF recruitment by blocking the binding of XPF to SLX4.

The data in this paper are interesting and of a high technical quality. The main concern in terms of suitability specifically for EMBO is that the conclusions are not particularly surprising or novel. For example, it has already been demonstrated that the ICL repair role of Xpf can be separated genetically from the NER role. This isn't a novel finding.

We agree with the referee that there are strong indications that the role of XPF in ICL repair can be separated from its role in NER, because several mutations have been identified that cause ICL sensitivity, but not UV sensitivity. We also say this clearly in the introduction of the manuscript. However, we think our work extends this finding in two major areas. First of all, we show biochemically that the mutations that cause ICL sensitivity are directly defective in ICL repair, but not in NER. Although this may be perceived as confirmatory, we strongly believe this is an important step in understanding the biochemical mechanism underlying this separation of function. In the second part of the manuscript we address this mechanistic question of why these mutants are selectively defective in ICL repair. This is important to understand the mechanism of ICL repair and the versatile nature of the endonuclease.

Also, cells deficient in Xpf or Slx4 show ICL repair defects; Xpf-deficient cells are NER-defective

but Slx4-deficient cells are not. So mutations in Xpf that prevent binding to Slx4 would be expected to inhibit ICL repair without affecting NER. This is not surprising or unanticipated.

It is true that SLX4 deficient cells are sensitive to ICLs, but not UV, indicating this could be an ICL repair specific factor. However, very little is known about the interaction of XPF and SLX4, only one XPF mutant has been reported to disrupt this interaction. We now show that this mutant (xLXPF^{G314E}) does not disrupt the interaction with full length SLX4 and is not defective in ICL repair. We also show that making a more drastic mutation (xLXPF^{ΔNSGW}) in the same area of XPF does inhibit repair, but still does not affect SLX4 interaction. Moreover, we now identified another XPF mutation (xLXPF^{L219R}) that does prevent this interaction. Our data supports a model in which there are two interaction sites between XPF and SLX4, a stable one required for ICL recruitment, and a transient one important for nuclease activation. This is a major advancement in our understanding of the how SLX4 regulates XPF-ERCC1 during ICL repair. We have made this more clear in the revised manuscript.

Similarly, the Knipscheer lab has already shown the role of Xpf in ICL repair is at the unhooking stage, so it's not surprising that the ICL repair-deficient mutants in Xpf are deficient in unhooking. That's the step Xpf catalyzes.

Yes, we have shown previously that absence of XPF-ERCC1 prevents ICL unhooking during repair. However, XPF-ERCC1 could have additional roles downstream in the HR step of ICL repair. Therefore, it is possible that certain mutations allow unhooking, but prevent these downstream functions. Here, we show definitively that all mutants we tested have a defect in unhooking incisions.

What would be novel would be some insight into why it is that the ICL repair-defective mutants do not affect NER.

In this study we have chosen to study why these mutants affect ICL repair, not why they do not affect NER. We felt this was the more interesting question to answer.

Getting some handle on the difference between Xpf in the two different contexts would represent an important advance. But this paper doesn't go there, and so in its current form its suitable for a more specialist journal (JBC?).

One other point: two mutants which do not bind to Slx4 are used - G314E and deltaNGWS. The former is proficient for ICL repair while the latter is not. The explanation given is that G314E doesn't disrupt the Slx4 interaction sufficiently to inhibit ICL repair. But there's no data on the difference between the two mutants in terms of Slx4 binding - they may be equally defective.

Using a pull down experiment after overexpression of SLX4 with XPF mutants in Sf9 insect cells we show that the XPF^{ΔNSGW} mutant still interacts normally with SLX4. We have now added data to figure 6B showing that this is also the case for the XPF^{G314E} mutant. However, based on previous reports (Andersen et al. Mol Cell 2009, Guervilly et al. Mol Cell 2015) and our finding that the XPF^{ΔNSGW} mutant is defective in ICL unhooking and repair, there is likely an important interaction between this region of XPF and SLX4. Mutating this interaction site does not prevent XPF – SLX4 interaction because we show there is a second, high affinity, interaction site involving the known SLX4 MLR domain and the leucine 219 of XPF.

If so, it's possible that the NGWS affects an aspect of Xpf function other than Slx4 interaction which explains the ICL repair defect.

This is a possibility we can not completely discard. However, since the XPF^{ΔNSGW} mutant complex has a mutation in the same residue that was previously shown to be required for the interaction with the BTB domain of SLX4 this is the most likely explanation. We have now addressed the possibility that it could also affect another aspect of XPF function in the discussion on page 16.

Its true that the L219R mutation inhibits Slx4 interaction and ICL repair but its simply correlation. Having an Xpf mutant that doesn't interact with Slx4 and that doesn't affect ICL repair argues the interaction with Slx4 is not important for ICL repair and this would need to be addressed.

We do not describe a mutant that does not interact with SLX4 and does not affect ICL repair. The only mutant that shows reduced SLX4 interaction is the L219R mutant and that mutant is defective in ICL repair. SLX4 is absolutely required for the recruitment of XPF to the ICL, this is shown in figure 6A. Therefore, it is highly unlikely that a mutant exists that does not interact with SLX4, but is competent in ICL repair.

Referee #3:

In a previous paper in Mol Cell in 2014, the authors showed that ERCC1-XPF is responsible for the unhooking step in interstrand crosslink (ICL) repair and that this activity depends on an interaction with SLX4. In the present paper, they analyze how a number of mutant alleles of XPF, isolated from FA patients and in screens in drosophila affect ICL repair and nucleotide excision repair (NER - XPF was first described as an NER protein).

The authors use their xenopus oocyte system to monitor replication-dependent ICL repair and expressed and purified six mutant XPF proteins. Consistent with the patient and cellular phenotypes of these proteins, they are proficient in NER, while displaying a defect in ICL repair. Importantly, the authors can pinpoint the ICL repair defect to the unhooking step, answering a long-standing question in the field, whether the key role of XPF in ICL repair is in the unhooking or a later step in recombination. Interestingly, with the exception of the L219P mutant all of the proteins are recruited to sites of ICL repair and interact with SLX4, showing that their defect in ICL repair results from a defect in positioning at the incision site, either through a defect in the interaction with DNA around the nuclease active site (R670S, S767F), or a defect due positioning in the ICL repair complex likely by protein-protein interaction (C225R, ΔNSGW). By contrast L219P does not bind SLX4 and is not recruited to ICL repair sites.

This work is carried out to a high technical standard and the great care that was taken to ensure that none of the defects observed was due to protein aggregation, a common problem with XPF mutant proteins, is particularly impressive. The work significantly adds to our understanding of the mechanisms by which XPF contributes to ICL repair.

A few minor concerns should be addressed prior to publication:

1) p3. line 16: It would be worth mentioning that all XP patients with residual NER activity have a relatively mild phenotype, due to residual levels of XPF protein. This low levels of XPF proteins are believed to sufficient to support ICL repair as the patients do not suffer from FA (Ahmad et al, PLoS Genet, 2010, 6, e1000871)

This is a valid point and we have added it more clearly to the introduction on page 4.

2) p4. bottom. The ability of ERCC1-XPF to incise both sides of an ICL was first shown by Wood and coworkers (Kuraoka et al, JBC, 2000, 275, 26632). This reference should be cited here.

The reviewer is absolutely right and we should have included it in the initial manuscript. However, due to drastic shortening of the introduction as requested by reviewer #4 this part is no longer in the revised manuscript.

3) p6. line 5: Ahmad et al, PLoS Genet, 2010, 6, e1000871 should be cited as the reference characterizing the cellular status of XPF in XP-F patients.

We have added this reference to page 4.

4) P12. bottom: Bergstralh and Sekelsky, Trends Genet 2008, 24, 70 would be good to cite for a discussion of how XPF might function downstream of incision in ICL repair.

We thank the referee for the suggestion and have added the reference to page 11.

5) P14. line 10 - An alternative explanation for why XPF-ΔNSGW is defective in ICL repair is that

it may well have an interaction defect with SLX4, but that this defect does not affect recruitment to sites of ICLs, but rather the position of the two proteins at the sites of ICL incision site. It is entirely possible that there are multiple sites of interaction between SLX4 and XPF. This point should be considered here, on p.15 (line 10) and in the discussion on p.16/17.

This is exactly the mechanism we tried to put forward in the discussion. We have now added additional data that corroborates with this model and have moved the explanation to the result section. In addition, we have added a model to Fig 7. We show that an SLX4 mutant that lacks the MLR domain does no longer interact with XPF, which is consistent with experiments in human cells. This indicates that a high affinity binding site between the MLR domain of SLX4 and leucine 219 of XPF is required for recruitment of XPF to the lesion. Furthermore, we purified the isolated BTB domain of SLX4 and showed that this does not interact with high affinity with XPF. However, based on previous reports (Andersen et al. Mol Cell 2009, Guervilly et al. Mol Cell 2015) and our finding that the XPF^{ΔNSGW} mutant is defective in ICL unhooking and repair, there is likely an important, but transient interaction between this region of XPF and SLX4 required for nuclease activation possibly by proper positioning.

Referee #4:

In this paper Knipscheer explores the role of the excision nuclease Xpf-Ercc1 in replication coupled DNA crosslink repair. Building on her own pioneering work with J Walter where together they established how a DNA crosslink is repaired during DNA replication using a *Xenopus* egg extract cell free system. In later follow up work Knipscheer established how the key excision or unhooking step occurs through the interplay between the FA pathway and Slx4 with Xpf-Ercc1 (Molecular Cell 2014). Now in this paper they carry out a more detailed analysis of the nuclease complex with Slx4, more specifically they exploit single mutations in the Xpf/Ercc1 complex that co-segregate with a human syndrome with overlap with FA and Cockayne syndrome. These mutations cause DNA crosslinker sensitivity but appear not to impact on the role of this complex in nucleotide excision repair.

The main conclusions of this current study is that mutants of Xpf-Ercc1 complex can be divided into two groups - those that impact both NER and CX repair (nuclease active site mutants), those that work in NER but do not unhook DNA crosslinks. This latter group falls into two further groups - those that bind Slx4 and are then not recruited to the crosslink, and those that bind Slx4 are recruited to the CX but fail to unhook it. They correctly conclude from this work that Slx4/ Xpf-Ercc1 complex not only works to recruit the nuclease to the damage site but also must somehow position the nuclease once it is there so that it cuts this substrate.

These are novel and important insights into DNA CX repair and therefore merits very strong consideration for publication in EMBO. However despite my obvious enthusiasm for the work I would like the authors to address my following concerns.

1. Generally the manuscript is written rather poorly, the introduction is way too long and utterly boring to read. After several espresso cups I did manage to wade through this tome akin to reading War and Peace in one setting for a romantically disinclined individual. It should be 1/3 the size and can be much more succinct.

We regret to hear the referee required a high dose of caffeine to get through our introduction. We have now drastically shortened it and hope he/she can now get through on a cup of tea.

2. Generally the figures are just appallingly laid out. I use the test that one should be able to "read" the figures without having to look at the text. I found this impossible in this case. The figures in her and Walters papers are really excellent - she should consider emulating these or at the very least matching their quality and clarity.

We have made many changes to improve the layout of the figures and added better cartoons for clarification.

3. The last figure should have a model that sums up what they are saying in the paper. Without such

a visual encapsulation of the message its impact will be lost to many.

This is an excellent suggestion and we have now added a model to Fig 7.

4. Fig 1 is excellent then all this goes rather downhill. Fig 2 WB should have a loading control. Perhaps space should be devoted to the Walter Knipsheer model and how this is assayed in their graphs.

All western blots on figure 2 contain samples taken straight from extract without any manipulations. We have never added loading controls to these blots in all previous papers (including the ones from Johannes Walter's lab). In these particular cases a loading control would not give extra information because we are merely showing the depletion (that we also confirm functionally) and the level of protein we add back.

The Walter/Knipscheer ICL repair model is now added to Figure EV1 and can be easily pulled up for clarity. In addition, we have clarified the cartoons explaining the assays in the main figures.

5. Figure 3 : Awful ! Have they shown that depleting other components of NER abrogates UDS in this assay ?

We thank the reviewer for this good suggestion. We have now depleted PCNA and XPA and show that this inhibits UDS in this assay. This data is added to Fig EV4.

2B why is the UDS analysis only provided for one of the mutants and not all of them shown in the panel below.

We have now added the data for all mutants to Fig 3.

2C there is no stats here and I am concerned about the range of intact excision repair, for instance some the mutants are even better than the wild type (the last one for instance).

We have statistically analyzed our data and show that the mutants are not statistically different from the wild type protein, except for the catalytically inactive XE^{D668A} mutant. This has been added to the manuscript.

I would also like to see this data corroborated by showing by slot blot that CPD dimers are removed (there are good antibodies against these lesions that track their removal. Getting this figure right matters since the rest of the paper builds on the mutants that work in NER but not in CX repair.

We have analyzed our plasmid with the antibody against CPD's. We did not use slot blots because they suffered from high nonspecific background signal but we used the same antibody in an ELISA assay (Cell Biolabs, USA). With a low UV dose we could clearly see that the CPD damage is repaired in Xenopus egg extract (Fig EV4C). For our UDS assay a higher UV dose was required to obtain enough signal to measure UDS unambiguously. At this high dose of UV we could not observe the decrease in CPDs, most likely because only a minor fraction is repaired. This data is added to Fig EV4C.

6. Figure 4 again difficult to follow- work in improving clarity here . Excessively cropped WB are not really acceptable these days, also again no loading control.

We have made changes to this figure to improve clarity and have reduced cropping of the blots.

7. Figure 6 no loading control.

See above

Thank you for submitting your revised manuscript on ICL repair-specific functions XPF-ERCC1 for our editorial consideration. Two of the original referees have now once more assessed the study, and I am pleased to inform you that they both consider the manuscript significantly improved and the key concerns satisfactorily addressed. We shall therefore be happy to accept the manuscript for publication in The EMBO Journal, pending correction of a few minor points noted by referee 3.

REFeree REPORTS

Referee #3:

The authors have satisfactorily addressed all the issues raised by the reviewers. I think even reviewer 4 will be able to relax now.

After addressing the minor concerns below the manuscript is now suitable for publication in EMBO J.

- 1) p3. Call XPF structure specific instead of flap endonuclease?
- 2) p4. XPF is suddenly referred to XPF (FANCD1) without stating that mutation in XPF cause FA. This is confusing and it would be better to state earlier in the paragraph something like: "consistent with a defect caused by XPF deficiency in ICL repair, mutations in XPF are also associated with FA" or then not refer to it as XPF (FANCD1) in this part of the text.
- 3) p8. 3rd paragraph: ...we used xenopus egg extracts (plural)
- 4) p12. Last paragraph: XPF leucine 219 is part of (not lysine!)

Referee #4:

The authors have greatly improved the text and addressed the majority and indeed my most prescient concerns. This paper deserves to be published in EMBOJ

Corresponding author: Puck Knipscheer

Manuscript Number: EMBOJ-2016-95223